# Robustness of anthropogenically forced decadal precipitation changes projected for the 21st century

Honghai Zhang [1,2] & Thomas L. Delworth[2]

Precipitation is characterized by substantial natural variability, including on regional and decadal scales. This relatively large variability poses a grand challenge in assessing the significance of anthropogenically forced precipitation changes. Here we use multiple large ensembles of climate change experiments to evaluate whether, on regional scales, anthropogenic changes in decadal precipitation mean state are distinguishable. Here, distinguishable means the anthropogenic change is outside the range expected from natural variability. Relative to the 1950–1999 period, simulated anthropogenic shifts in precipitation mean state for the 2000–2009 period are already distinguishable over 36–41% of the globe—primarily in high latitudes, eastern subtropical oceans, and the tropics. Anthropogenic forcing in future medium-to-high emission scenarios is projected to cause distinguishable shifts over 68–75% of the globe by 2050 and 86–88% by 2100. Our findings imply anthropogenic shifts in decadal-mean precipitation will exceed the bounds of natural variability over most of the planet within several decades.

[1] Program in Atmospheric and Oceanic Sciences, Princeton University, 300 Forrestal Road, Princeton, NJ 08544, USA. [2] Geophysical Fluid Dynamical Laboratory/NOAA, 201 Forrestal Road, Princeton, NJ 08540, USA. Correspondence and requests for materials should be addressed to H.Z. (email: honghaiz@princeton.edu) or to T.L.D. (email: tom.delworth@noaa.gov)

Decadal variability in precipitation can have drastic impacts on environment and society. For example, the notorious 1930s American Dust Bowl, a persistent decade-long drought, is one of the most devastating environmental catastrophes that have stricken the United States during the past century[1–3]. Projecting future decadal changes in precipitation—particularly those caused by anthropogenic activities—have been not only a longstanding goal for scientists but also of great interest to the public. Decadal changes in precipitation are dominated by internal climate variability (arising from natural processes internal to the climate system and their interactions)[4–11] that is inherently unpredictable beyond a decade (owing to the chaotic nature of the climate system)[7, 12–15]. This dominance of unpredictable internal climate variability (noise) presents an enormous challenge in the projection and assessment of anthropogenically caused decadal changes in precipitation (signal) owing to the weak signal-to-noise ratio. Nonetheless, projecting and assessing anthropogenic decadal changes in precipitation are of crucial importance for both scientific understanding and practical applications (e.g., policy making) associated with climate mitigation and adaptation.

In this work, we assess where and when regional-scale decadal shifts in precipitation mean state that arise from anthropogenic forcing can be robustly distinguished from the background of unpredictable low-frequency (longer than a decade) internal climate variability in future ensemble projections. We focus on the mean state because anthropogenic shifts in mean state are the most predictable component in climate change[16]. A shift in precipitation mean state is defined as distinguishable when the amplitude of the shift is outside the range of what could occur from low-frequency internal climate variability (see Methods for more details). Specifically, we address the following question: On a decade by decade and grid box by grid box basis, which projected shifts in precipitation mean state relative to the 1950–1999 climate can be distinguished from low-frequency internal climate variability and attributed to anthropogenic forcing? Answers to this question can inform policies on water resource management, agricultural development, and food and society security. To answer the above question, we examine the evolution of decadal shifts in precipitation mean state after calendar year 2000 (relative to the 1950–1999 climate), with a focus on projections up to calendar year 2050. As to the decadal shifts in precipitation mean state, we quantify the relative contributions from anthropogenic forcing and natural climate variability, the latter of which includes both unforced low-frequency internal climate variability and forced natural (e.g., volcanic) variability. We show that, on regional scales, the anthropogenically forced decadal shifts in precipitation mean state are becoming progressively more distinguishable from natural climate variability with each decade over more areas of the globe.

## Results

**Model set-up.** This analysis is enabled by a large set of simulations (about 21000 model years in total) from two climate models, the National Center for Atmospheric Research (NCAR) Community Earth System Model version 1 (CESM1)[17] and the Geophysical Fluid Dynamics Laboratory (GFDL) Forecast-oriented Low Ocean Resolution (FLOR) flux-adjusted model[18]. The simulations include four multi-millennial preindustrial control experiments and three state-of-the-art large model ensembles of climate change experiments (Table 1). The control simulations—with varying degrees of ocean–atmosphere coupling—are used to assess low-frequency internal climate variability, while the large model ensembles are used to estimate externally forced signals. Conducted using a single climate model by prescribing the same external forcing but different initial conditions, members of each large model ensemble are composed of the same externally forced signal and different internally generated variability. The average of each large model ensemble provides a much-refined estimation of the forced signal through the cancellation of random phases of internal variability present in the various ensemble members. Because of this unique advantage in estimating externally forced signals, the large model ensemble is especially suitable for extracting externally forced decadal changes in precipitation (dominated by internal climate variability).

The three large model ensembles include a 35-member ensemble (1921–2100) conducted with the NCAR CESM1, a 35-member ensemble and a 30-member ensemble (both 1941–2050) conducted with the GFDL FLOR. The two 35-member ensembles (termed ALLFORC) are driven by observationally based estimates of historical forcing before 2005 and the Representative Concentration Pathway (RCP) emission scenarios after 2005 (RCP 8.5 for CESM1 and RCP 4.5 for FLOR); the 30-member FLOR ensemble (termed NATURAL) is driven by natural-only (solar and volcanic) forcing before 2005 and solar-only forcing after 2005 (anthropogenic forcing held constant).

## Table 1 Model simulations used in this study

|  |  |  | Total number of model years analyzed |
|---|---|---|---|
| *GFDL FLOR ~0.5° atmosphere/land and ~1° sea ice/ocean* |  |  |  |
| Fully coupled control | Preindustrial forcing | 3500 years | 3400 (101–3500) |
| 30-member NATURAL | Natural historical forcing (solar variations, volcanos) before 2005; solar variability-only (quasi-11-year cycle) afterwards | 1941–2050 | 3030 (1950–2050) |
| 35-member ALLFORC | All historical forcing before 2005; RCP4.5 afterwards | 5 members: 1861–2100 30 members: 1941–2050 | 3535 (1950–2050) |
| *NCAR CESM1 ~1° atmosphere, land, sea ice and ocean* |  |  |  |
| Fully coupled control | Preindustrial forcing | 2200 years | 1801 (400–2200) |
| Atmosphere/land control | Preindustrial forcing | 2600 years | 2600 |
| Atmosphere/land/slab ocean control | Preindustrial forcing | 1000 years | 1000 |
| 35-member ALLFORC | All historical forcing before 2005; RCP8.5 afterwards | 1 member: 1850–2100 34 members: 1920–2100 | 5285 (1950–2100) |

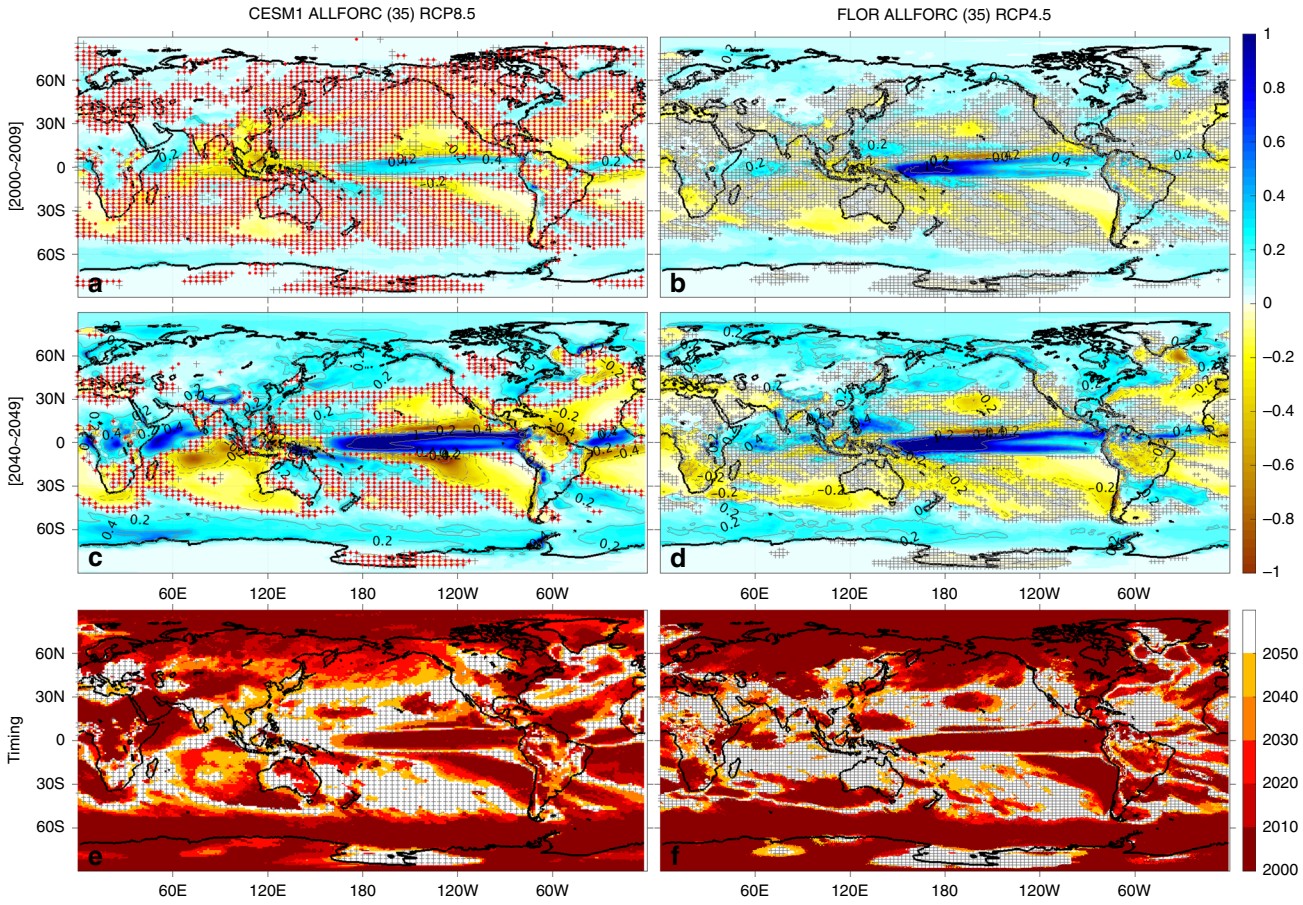

**Fig. 1** Decadal evolution of annual precipitation mean state and timing of distinguishability in precipitation changes. **a–d** decadal evolution of changes in annual precipitation mean state (ensemble average, shading, mm day$^{-1}$) relative to the 1950–1999 climate during 2000s (**a, b**) and 2040 s (**c, d**) in CESM1 ALLFORC RCP8.5 (**a, c**) and FLOR ALLFORC RCP4.5 (**b, d**). Time goes down, as denoted to the left of the figure (changes during 2010s, 2020s and 2030s are shown in Supplementary Figure 3). Contours at intervals of 0.2 mm day$^{-1}$ are labeled in gray, with dashes denoting negative precipitation changes. Gray crosses in both columns denote that changes in precipitation mean state are not distinguishable from internal climate variability estimated from fully coupled control simulations (see Methods for details on the distinguishability test); red stippling in (**a, c**) denotes changes in precipitation mean state are not distinguishable from internal climate variability estimated from the atmosphere/land-only control simulation of CESM1. **e, f** timing of distinguishable changes in precipitation mean state in CESM1 (**e**) and FLOR (**f**), defined as the first decade when precipitation changes become distinguishable and remain so thereafter. The gray crosses in (**e, f**) means no distinguishability by 2050

Comparing the ALLFORC and NATURAL ensembles can help further estimate the anthropogenically forced signals.

Both models simulate historical precipitation changes over recent decades that are consistent with observations over most of the global land (Supplementary Figures 1 and 2), thereby increasing our confidence in the utility of these simulations for assessing whether anthropogenic decadal changes in precipitation mean state in future projections are distinguishable from natural variability. For each decade after 2000, the mean state is defined as the ensemble average within each large model ensemble over 10*$N$ model years, where $N$ is the ensemble size (10 years from each ensemble member). For example, in a 35-member ensemble, the decadal mean state for 2030–2039 is calculated as the numerical average over 350 model years. The relatively large model ensembles allow a more robust estimate of the mean state for each decade.

**Near-term projections**. We analyze the distinguishability of anthropogenic decadal shifts in precipitation mean state averaged over five periods: January-to-December (annual), November-to-April (NDJFMA), May-to-October (MJJASO), December-to-

February (DJF) and June-to-August (JJA), separately. Besides precipitation, we have also conducted the same analysis for precipitation minus evaporation (PmE)—the net water flux at the surface. We have found similar results for five periods and also for PmE. Here we focus on the annual mean precipitation and discuss the differences from the seasons (NDJFMA, MJJASO, DJF, and JJA) and PmE as necessary.

The decadal evolution of projected shifts in annual precipitation mean state (computed as the ensemble average) during 2000–2050 (relative to the 1950–1999 reference period) is similar on the global scale between the CESM1 and FLOR ALLFORC ensembles (Fig. 1 and Supplementary Figure 3). In response to future anthropogenic warming (RCP 8.5 and 4.5), the global precipitation mean state features a moistening trend in high latitudes and tropics and a drying trend in subtropics to middle latitudes. This global pattern has been largely explained by the "wet get wetter, dry get drier" mechanism[19]: with the increase in atmospheric moisture, large-scale atmospheric circulation will converge more moisture into regions of climatological convergence (wet get wetter), and diverge more moisture from regions of climatological divergence (dry get drier). Other mechanisms have been proposed in the tropics and subtropics. Over the

**Table 2 Fraction (%) of the global area (weighted by latitudes) with distinguishable shifts in annual/NDJFMA/MJJASO precipitation mean state, respectively**

| | GFDL FLOR 35-mem ALLFORC RCP4.5 | | | NCAR CESM1 35-mem ALLFORC RCP8.5 | | |
|---|---|---|---|---|---|---|
| | Land | Ocean | Total | Land | Ocean | Total |
| 2000–2009 | 42.8/40.0/34.2 | 40.8/33.5/39.8 | 41.4/35.3/38.2 | 40.3/29.8/29.1 | 33.2/24.0/32.4 | 35.5/25.9/31.3 |
| 2010–2019 | 54.5/49.5/47.8 | 49.2/41.8/48.5 | 50.7/44.0/48.3 | 55.7/48.8/46.2 | 43.4/35.6/44.1 | 47.4/39.9/44.8 |
| 2020–2029 | 59.9/56.6/55.3 | 58.8/52.0/57.2 | 59.1/53.4/56.7 | 65.6/60.7/56.7 | 56.1/49.4/54.6 | 59.2/53.1/55.2 |
| 2030–2039 | 65.8/63.2/59.9 | 62.3/56.8/61.7 | 63.3/58.7/61.2 | 73.5/69.7/63.0 | 63.7/58.3/64.8 | 66.9/62.0/64.2 |
| 2040–2049 | 70.8/66.0/65.1 | 66.7/60.8/67.0 | 67.9/62.3/66.4 | 79.0/75.1/69.7 | 72.4/65.6/73.2 | 74.5/68.7/72.1 |
| 2050–2059 | | | | 84.2/80.6/74.5 | 77.6/72.3/79.2 | 79.8/75.0/77.6 |
| 2060–2069 | | | | 86.4/84.2/79.0 | 80.8/78.0/83.8 | 82.6/80.0/82.2 |
| 2070–2079 | | | | 88.3/86.9/82.3 | 84.1/81.1/85.5 | 85.5/83.0/84.4 |
| 2080–2089 | | | | 89.1/87.9/83.3 | 86.3/84.2/87.4 | 87.2/85.4/86.1 |
| 2090–2099 | | | | 90.0/89.8/84.1 | 87.1/84.9/88.6 | 88.1/86.5/87.1 |

The distinguishability of externally forced shifts in precipitation mean state (relative to the 1950–1999 mean climate) is estimated against internal climate variability in fully coupled control simulations (see Methods for more information). Results are shown for the two ALLFORC ensembles.

tropical oceans, precipitation changes follow the pattern of sea surface temperature changes owing to its control on tropospheric moist instability, with "warmer get wetter"[20]; in the subtropics, precipitation response is driven primarily by the fast adjustment to $CO_2$ forcing, including land-sea warming contrast and direct $CO_2$ radiative forcing[21], with a secondary contribution from the slow poleward expansion of the Hadley Cell and subtropical dry zone[22, 23]. All these mechanisms imply that these decadal shifts in precipitation mean state result from anthropogenic forcing. However, these decadal shifts can also be caused by low-frequency (longer than a decade) internal climate variability. So, where and when can the projected decadal shifts be distinguished from internal climate variability and attributed to anthropogenic forcing?

During the 2000s (i.e., 2000–2009) in both ALLFORC ensembles (Fig. 1a–b), simulated shifts in precipitation mean state are not distinguishable from low-frequency internal climate variability over most of the globe. However, in certain regions including the Arctic, Southern Ocean, eastern subtropical oceans, equatorial Pacific and a few land regions over Eurasia, North/South America and Antarctic, distinguishable shifts have already emerged in the simulations. As expected, the area of distinguishable shifts in precipitation mean state increases with radiative forcing. This increase occurs mainly over higher latitudes (poleward of ~40°) and eastern subtropical ocean basins. By 2050, both models project distinguishable changes in precipitation mean state over more than 65% of the globe (about 68% in FLOR with 71% over land; 75% in CESM1 with 79% over land) (Table 2). Without anthropogenic forcing, the FLOR NATURAL ensemble (Supplementary Figure 3c) projects distinguishable changes in precipitation mean state only in a few scattered regions (including parts of the Arctic and the Southern Ocean), which account for about 13% of the global area by 2050 (Fig. 2). These distinguishable changes in the NATURAL ensemble are likely associated with a small warming trend in future projections owing to the difference in the imposed volcanic forcing between the 1950 and 1999 reference period (observed volcanic emissions) and the period after 2005 (no volcanic activity). The large difference between ALLFORC and NATURAL ensembles indicates that most of the distinguishable shifts in annual precipitation mean state are attributable to anthropogenic forcing.

To quantify the time of emergence of anthropogenic shifts in annual precipitation mean state (relative to the 1950–1999 period), we compute the decade when projected precipitation changes first become distinguishable from natural climate variability and remain so thereafter for at least two decades (see

Methods for more details). The CESM1 and FLOR ALLFORC ensembles project a similar global pattern of the timing (Fig. 1e, f). The earliest distinguishable signals (including the moistening over high latitudes and equatorial Pacific and the drying over part of subtropical oceans) are simulated during the 2000s over about 36–41% of the globe, with 43% of global land in FLOR and 40% in CESM1 (Table 2 and Fig. 2). Over Eurasia and North America north of about 40°N, southeast Asian monsoon region and most of South America, both models project distinguishable shifts in precipitation mean state during the decades between 2000 and 2040. By 2040, around 65% of the globe (~63% in FLOR and 67% in CESM1) is projected to experience distinguishable shifts in precipitation mean state (Table 2 and Fig. 2). Despite its stronger RCP forcing, CESM1 in general projects a later timing than FLOR, because it simulates a smaller ratio between the signal of changes in precipitation mean state and the noise of internal climate variability during the early decades of the 21st century (Fig. 3a, b). However, the rate of increase in the area of distinguishable precipitation shifts is larger in CESM1 (39%/50 years) than in FLOR (27%/50years) (Fig. 2), consistent with the stronger RCP forcing in CESM1.

As is the case for annual mean, the projected shifts in precipitation mean state for seasonal means (NDJFMA and MJJASO in Supplementary Figures 4 and 5, respectively) are similar between the CESM1 and FLOR ALLFORC ensembles. Furthermore, the projected shifts for seasonal means also resemble those for annual mean (see Fig. 1), with the global-scale "wet get wetter, dry get drier" pattern (i.e., a general moistening over high latitudes and tropical oceans and a general drying over subtropics and middle latitudes). However, noticeable differences exist among the seasonal and annual results. Overall on the global scale, following the seasonal migration of the precipitation climatology, the global meridional moistening-drying alternating pattern shifts slightly northward in MJJASO compared to NDJFMA (Supplementary Figures 4 vs 5). This geographical northward shift of precipitation changes leads to contrasting trends between MJJASO and NDJFMA and the resultant weak annual trend over regions near the moistening-drying transition zone. For example, over central-western North America and central-southwestern Eurasia, a drying trend is projected during MJJASO (Supplementary Figure 5), in contrast to the moistening trend during NDJFMA (Supplementary Figure 4); as a result, the annual mean trend (Fig. 1) over these regions is relatively weak. In addition, over monsoon regions such as southeast Asia and southeastern South America, the seasonal precipitation shifts tend to show opposite signs between

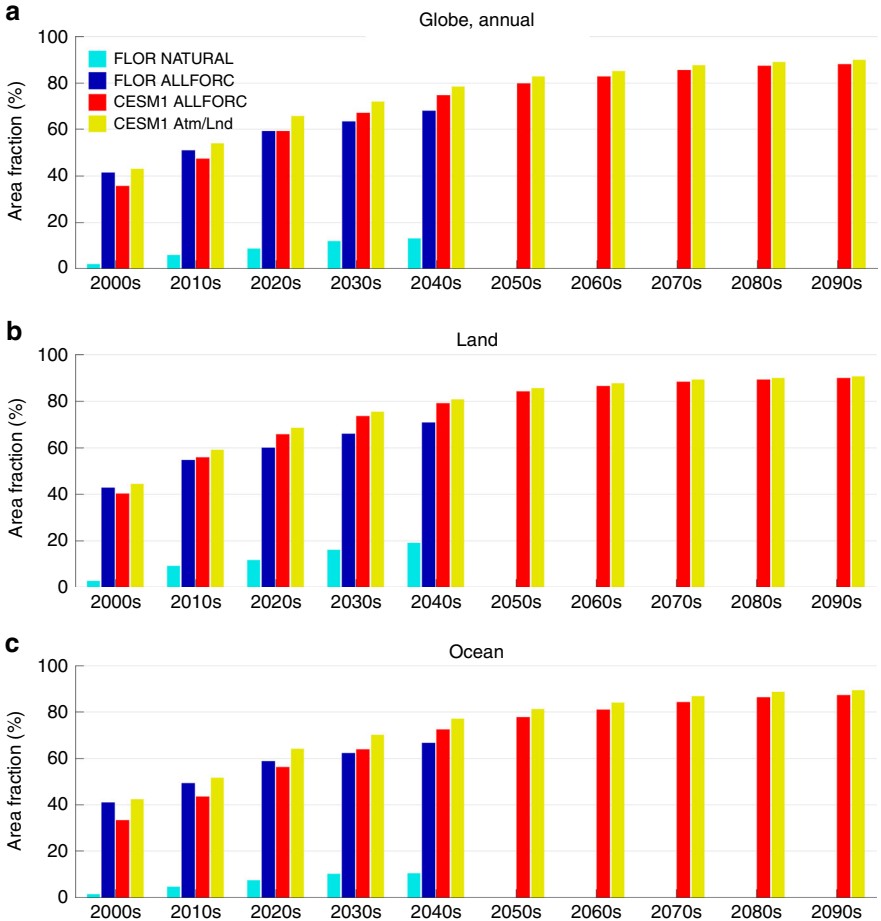

**Fig. 2** Fraction of area with distinguishable changes in annual precipitation mean state. **a** total global area, **b** global land, and **c** global ocean. Cyan, blue, and red bars show fraction of area with distinguishable changes against fully coupled internal climate variability in FLOR NATURAL, ALLFORC, and CESM1 ALLFORC, respectively, while yellow bars denote that in CESM1 ALLFORC against atmosphere/land-only internal climate variability

NDJFMA and MJJASO, and the annual precipitation shifts are dominated by the shifts during the monsoon season (e.g., MJJASO for southeast Asia and NDJFMA for southeastern South America).

The distinguishability of the anthropogenic shifts in precipitation mean state for both NDJFMA and MJJASO (also DJF and JJA, not shown) exhibits a global pattern similar to that for the annual mean state (compare Fig. 1 with Supplementary Figures 4 and 5), with the earliest distinguishable shifts emerging in high latitudes, eastern subtropical oceans and equatorial Pacific. However, a close comparison reveals that the fraction of the globe with distinguishable anthropogenic shifts is slightly larger for the annual precipitation mean state than for the seasonal precipitation mean state, especially during early decades (by about 5–10% larger than NDJFMA and 2–4% than MJJASO during 2000–2050, see Table 2). This earlier distinguishability for the annual precipitation mean state is consistent with its overall stronger signal-to-noise ratio during early decades (compare Fig. 3a, b, Supplementary Figures 6a, b and 7a, b), which arises mainly from the difference in the noise. While the signal for the annual precipitation mean state is simply the average between those for NDJFMA and MJJASO, the annual noise is weaker than the average between the two seasonal counterparts over most of the globe (Fig. 4), leading to the overall stronger signal-to-noise ratio for the annual precipitation mean state during early decades. A similar difference is identified between the 6-month (NDJFMA and MJJASO) and 3-month (DJF and JJA) seasonal means, with

the latter having a slightly stronger noise and weaker distinguishability (not shown). Note that this difference in signal-to-noise ratio gets smaller as the signal grows with time (Table 2).

Despite the overall earlier distinguishability of anthropogenic precipitation shifts in annual vs seasonal means, there are different stories over certain regions. For example, over middle latitudes (e.g., central-western North America), opposite precipitation shifts of similar magnitudes are projected between NDJFMA and MJJASO; as a result, anthropogenic shifts in the annual precipitation mean state are very weak and thus less distinguishable. Over some monsoon regions such as southeast Asia and southeastern South America, the annual precipitation shifts and their distinguishability are dominated by (and slightly weaker than) the monsoon season. Specifically, the annual moistening over southeast Asia follows MJJASO (and JJA) and is distinguishable during 2000–2040 (about 2040s in CESM1 and 2000–2030 in FLOR), while over southeastern South America the annual moistening follows NDJFMA (and DJF) and is distinguishable during 2000–2030 (also slightly earlier in FLOR than CESM1). Note that over southeast Asia for the MJJASO (and annual) precipitation mean state (Fig. 1 and Supplementary Figure 5), FLOR projects a continuous moistening that becomes distinguishable around 2000–2030, while CESM1 first projects a distinguishable drying during the 2000s and then a distinguishable moistening during the 2040s. The drying in CESM1 is likely caused by anthropogenic aerosol forcing (cooling) that peaks around 2010[24], and the subsequent moistening is due to the

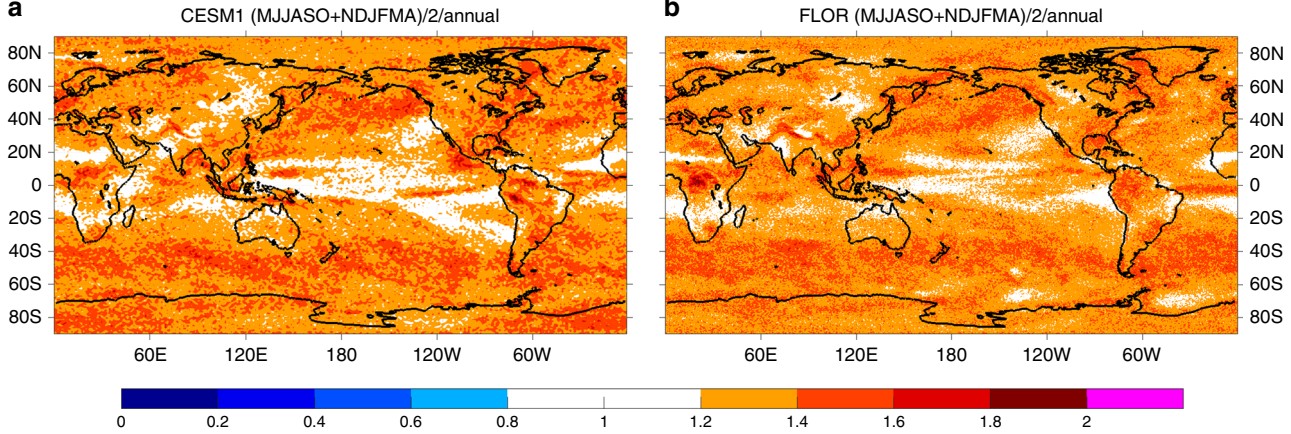

**Fig. 3** Signal-to-noise ratio and range of the noise for annual precipitation. **a–d** signal-to-noise ratio and **e, f**: range of the noise (mm day$^{-1}$) for CESM1 (**a, c, e**) and FLOR (**b, d, f**), respectively. The signal is the ensemble-mean precipitation change shown in Fig. 1a–d and the noise is the annual precipitation low-frequency internal variability against which the signal is tested. The range of the noise is estimated as the difference between the maximum and minimum (i.e., the most positive and negative) values of the 5000 samples constructed with the Monte Carlo approach from fully coupled control simulations, where each sample represents the synthetic ensemble-mean precipitation change arising entirely from internal climate variability (see Methods for more details). The signal-to-noise ratio is computed as the ratio of positive (negative) ensemble-mean precipitation change to the maximum (minimum) value of the 5000 samples, which follows the definition of distinguishability (see Methods). Regions without distinguishability (i.e., signal-to-noise ratio between 1 and -1) are indicated by white color with gray stippling in (a-d) (showing the decades of 2000–2009 and 2040–2049, respectively)

**Fig. 4** Ratio of the average precipitation noise between NDJFMA and MJJASO to the annual precipitation noise. **a** CESM1 and **b** FLOR. The noise refers to the precipitation low-frequency internal variability and is estimated as the difference between the maximum and minimum (i.e., the most positive and negative) values of the 5000 samples constructed with the Monte Carlo approach from fully coupled control simulations. The annual noise is plotted in Fig. 3e, f while the NDJFMA and MJJASO noise is plotted in Supplementary Figures 6e, f and 7e, f, respectively

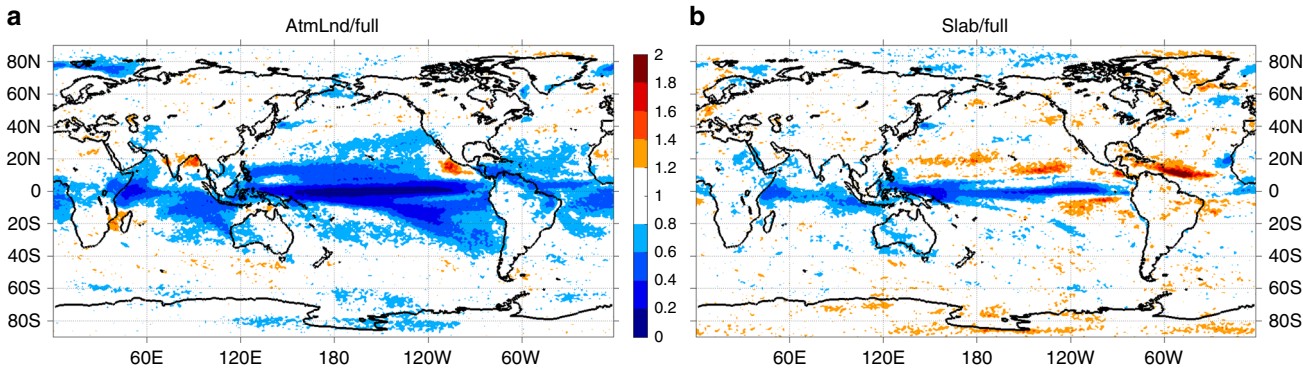

**Fig. 5** Ratio of annual precipitation low-frequency internal variability among the CESM1 preindustrial control simulations. **a** atmosphere/land-only vs fully coupled simulation and **b** atmosphere/land/slab ocean vs fully coupled simulation. The amplitude of precipitation low-frequency internal variability is estimated as the range (i.e., the difference between the maximum and minimum values) of the 5000 samples constructed with the Monte Carlo approach in each control simulation, where each sample represents the synthetic ensemble-mean precipitation change arising entirely from internal climate variability (see Methods for more details)

continuously increasing radiative forcing associated presumably with greenhouse gases[25–28] (as in FLOR).

Besides precipitation, we have also repeated the same distinguishability analysis for PmE—the net water flux at the surface. In both models, projected shifts in PmE mean state show very similar evolution to those in precipitation mean state on the global scale, and their distinguishability is also similar (Supplementary Figure 8 vs Fig. 1 for annual, Supplementary Figure 9 vs 4 for NDJFMA, Supplementary Figure 10 vs 5 for MJJASO; Supplementary Table 1 vs Table 2). This similarity suggests that changes in PmE are largely dominated by precipitation instead of evaporation. However, there are some land regions that exhibit notable differences between the projected shifts in precipitation and PmE, especially during MJJASO. For example, over central North America and central-western Eurasia during MJJASO, projected shifts in precipitation mean state (Supplementary Figure 5) are mostly weak and indistinguishable from internal climate variability by 2050, but projected shifts in PmE mean state (Supplementary Figure 10) show broad future drying that is distinguishable as early as 2000–2020. With weak changes in precipitation, the PmE drying during MJJASO arises from a stronger increase in evaporation, which is likely supported by net water storage from the PmE moistening during NDJFMA (see Supplementary Figure 9). Because of the opposite shifts between MJJASO and NDJFMA, the projected shifts in annual PmE mean state are rather weak and hardly distinguishable over these broad land regions (Supplementary Figure 8). As a result, the fraction of the global area with distinguishable shifts in the annual PmE mean state is not noticeably larger than that in the seasonal mean state during early decades (Supplementary Table 1) despite the overall weaker noise for annual PmE (Supplementary Figure 11), in contrast to the case for precipitation. Nonetheless, this relationship among the annual and seasonal distinguishability for PmE is still consistent with the associated signal-to-noise ratio (not shown).

**Long-term projections**. For CESM1 in response to the RCP8.5 emission scenario, the projected changes in precipitation mean state after 2050 exhibit a similar spatial pattern to that during the 2040s, but increase in magnitude with the imposed radiative forcing[29] (Supplementary Figure 12). By 2100, distinguishable shifts in precipitation mean state are projected over about 86–88% of the globe (Table 2 and Fig. 2). The small fraction of the area with indistinguishable shifts is primarily the transition zone between moistening and drying, where the anthropogenic

changes are weak. Similar results are also found for PmE (not shown).

**Origin of precipitation low-frequency internal variability**. we finally investigate the origin of precipitation low-frequency internal variability. Repeating the distinguishability analysis with the CESM1 atmosphere/land-only control simulation with fixed boundary conditions (as opposed to the fully coupled simulation, see Methods for more details), we find very similar results (Fig. 1a, c, Fig. 2, and Supplementary Figures 3a, c, 4a, c and 5a, c). A direct comparison of precipitation low-frequency internal variability between the two control simulations shows that precipitation internal variability has comparable magnitudes over most of the globe except for the tropics and subtropics (Fig. 5a). This result suggests that over most land areas and middle-to-high latitude oceans the variance of precipitation low-frequency internal variability arises primarily from internal dynamics of the atmosphere and land–atmosphere interactions, while in the tropics and subtropics ocean dynamics contribute substantially (e.g., > 60% in the tropical Pacific)[30]. A further comparison between the fully coupled and the slab ocean (with fixed ocean heat transport and no active ocean dynamics) control simulations (Fig. 5b) reveals that ocean dynamics in the tropics and subtropics can amplify precipitation low-frequency internal variability mainly through ocean–atmosphere thermodynamic coupling (i.e., heat and moisture exchange), but in the equatorial Pacific, ocean–atmosphere dynamical coupling (i.e., momentum exchange) is also required. These results highlight that precipitation low-frequency internal variability has very limited predictability (even in the tropics and subtropics) owing to the control of atmospheric internal dynamics and land–atmosphere interactions and can impart large uncertainties to the near-term (up to 50 years[5, 6] or even longer) projections of precipitation. However, we use large model ensembles to demonstrate that, despite these large uncertainties in total precipitation changes, anthropogenic shifts in precipitation mean state can be distinguished from natural climate variability over most of the globe by the middle of the current century.

**Discussion**

In our simulations, anthropogenic shifts in precipitation mean state are already distinguishable from natural climate variability over about 36–41% of the globe during the 2000s relative to the 1950–1999 climate. These earliest distinguishable signals include

a moistening over high latitudes and equatorial Pacific and a drying over eastern ocean basins near subtropical Highs. Over high latitudes and eastern subtropical oceans, the earliest distinguishability arises mainly from the weak internal climate variability (so that even small signals can readily emerge), while in the equatorial Pacific it is primarily due to the strong moistening signal.

In the tropics, the early emergence of an anthropogenic signal in precipitation mean state is similar to that in surface temperature reported in a number of previous studies[31, 32], but for a different reason. The early emergence of anthropogenic warming in the tropics results primarily from the relatively weak internal variability in surface temperature, while the early emergence of anthropogenic moistening there arises mainly from the relatively strong signal (since the associated internal variability is strong). Over high latitudes (particularly the northern hemisphere), the early emergence of an anthropogenic signal is projected for precipitation, but not for surface temperature. This difference is also due to the relative amplitude of internal climate variability: the strong internal variability in surface temperature delays the emergence of anthropogenic warming (despite the well-known polar amplification).

Our findings highlight the substantial impacts on future precipitation mean state that result from anthropogenic emissions. On the global scale, more than 60% and 85% of the globe is projected to experience distinguishable anthropogenic shifts in precipitation mean state by 2050 and 2100, respectively. On regional scales, anthropogenic signals with early distinguishability are projected for several land regions, such as the moistening over southeastern South America[10], northeastern North America and northern Eurasia during boreal cold seasons and the moistening over southeast Asia and the drying (in PmE) over central North America and central-western Eurasia during boreal warm seasons. These anthropogenically forced changes in precipitation mean state suggest that future hydrological extremes will occur around a shifted mean state that can intensify their strength (e.g., floods/droughts over wetter/drier mean state), therefore presenting severe challenges for both ecological and social systems[33, 34].

## Methods

**Models and experiments.** The three state-of-the-art large ensembles analyzed here are conducted with two global coupled climate models, the FLOR flux-adjusted model[18] developed at the GFDL and the CESM1[35] developed at the NCAR. The flux-adjusted FLOR employs fixed anomalous enthalpy, momentum, and fresh water fluxes to the ocean to bring its long-term climatology of sea surface temperature and wind stress closer towards observational estimates over 1979–2012; as a result, the climatology of other fields including precipitation and the atmospheric teleconnections have also been improved[10, 18, 36]. The GFDL FLOR has a horizontal resolution of approximately 0.5° for the atmosphere and land components and 1° (telescoping to 0.333° near the equator) for the ocean and sea ice components, while the NCAR CESM1 has a nominal horizontal resolution of 1° for all model components (atmosphere, ocean, land and sea ice). Note that both models have much finer spatial resolution than those used in some previously published large model ensembles (about 2–3°)[5, 6].

Two of the three large ensembles are performed with the FLOR, flux-adjusted version. The first ensemble has 35 members driven by identical observed estimates of both anthropogenic and natural forcing before 2005 and RCP 4.5 emission scenario[37] thereafter (termed ALLFORC), while the second ensemble has 30 members driven by identical observed estimates of natural-only (volcanic and solar) forcing before 2005 and solar-only forcing (quasi-11-year cycle) thereafter (anthropogenic forcing fixed at the 1941 level) (termed NATURAL). Both ensembles simulate the period from 1941 to 2050, except for five ALLFORC members covering 1861–2100. Members within each ensemble start from different conditions that are briefly described as follows: the five long members start from different years of a 3500-year preindustrial control simulation (Year 101, 141, 181, 221, and 261); the remaining 30 ALLFORC members and the 30 NATURAL members share the same initial conditions that are created by shuffling the 1940- and 1942-atmosphere/land states and 1941-ocean/sea ice states of the five long members: the first (second, third) 10 members start from the 1941-ocean/sea ice state of the first (second, third) long member combined with the 1940- or 1942-

atmosphere/land states of the five long members. We allow the model to adjust to the new initial conditions for the period 1941–1950, and only analyze model output after 1950.

The CESM large ensemble analyzed here has 35 members[17]. It simulates the period of 1921–2100, except for one member starting in 1850. All members are subject to identical observed estimates of historical (anthropogenic and natural) forcing before 2005 and RCP8.5 emission scenario[37] thereafter, but differ in their initial conditions (also termed ALLFORC; note its stronger RCP radiative forcing than FLOR ALLFORC). The long member (1850–2100) is initialized from year 402 of a 2300-year preindustrial control simulation, while the remaining members branch off from the long member in year 1921 with round-off level differences added only to the air temperature field. Consistent with the FLOR ensembles, we only analyze model output after 1950.

Besides the three large ensembles of the climate change experiment, four preindustrial control simulations are also analyzed. These simulations include the aforementioned 3500-year FLOR and 2300-year CESM1 fully coupled control runs, a 2600-year CESM1 control run only using its active atmosphere/land components driven by fixed boundary conditions (monthly mean sea surface temperature and sea ice averaged over years 402–1510 of the fully coupled control run) and a 1000-year CESM1 control run using its active atmosphere/land components coupled thermodynamically (via heat and moisture exchange, no momentum exchange) with a slab ocean. The slab ocean has a temporally constant but spatially varying depth that is set to be the climatological ocean mixed layer depth in the fully coupled control run. In addition, a heat flux adjustment, derived also from the fully coupled control run[38], is added to the slab ocean to compensate for the lack of active ocean dynamics; this heat flux adjustment varies only from month to month (not year to year) and is employed to drive the slab ocean surface temperature climatology towards that in the fully coupled control run. These long control simulations exhibit more or less climate drifts owing to models' radiative imbalance. The drifts are estimated using a low-pass filter with a cutoff period of 200 years and subtracted from both the control simulations and the corresponding large ensembles. Although we find that removing the model drifts does not affect our conclusions, here we present the results with model drifts removed. These long control simulations will be used to estimate low-frequency (decadal and longer) internal climate variability.

**Model evaluation.** The performance of the two models in simulating precipitation changes is evaluated against two observational products of global land precipitation, the Climate Research Unit at the University of East Anglia, version 3.24.01[39] and the Global Precipitation Climatology Centre dataset, version 7[40], both at 0.5° resolution. Here we compute precipitation changes between the last 10 years (1996–2005) and the first 46 years (1950–1995) during 1950–2005, and compare observations with individual members of the two ALLFORC ensembles. All model outputs are interpolated (with a globally conservative remapping method) onto the observational grid before the comparison. If the observed change in one grid box is within the range of those simulated by the 35 ALLFORC ensemble members, we consider that the model is consistent with observations in that grid box. The evaluation is performed for annual, NDJFMA, and MJJASO (see Supplementary Figures 1 and 2).

**Distinguishability.** We refer to a change in ensemble-mean, decadal-mean precipitation as "distinguishable" if the change lies outside the bounds of what one would expect from internal climate variability as estimated from multi-millennia preindustrial control simulations. Internal climate variability is estimated with a grid-scale Monte Carlo approach. For each control simulation, we do the following: at each grid point we first randomly select a 10-year period (to mimic any decade in the period 2000–2050) and a second non-overlapping 50-year period (to mimic 1950–1999), and then compute the difference between the time-mean of the 10-year period and the time-mean of the 50-year period—this difference results only from internal climate variability. We repeat this $N$ times (the value of $N$ is described below) to form the grand ensemble of these differences and compute the ensemble average. We then repeat the above process 5000 times to create a distribution of such ensemble-mean differences that could occur simply by chance from internal climate variability. We use $N = 30$ when evaluating the projected changes in the 30-member NATURAL ensemble, and $N = 35$ when evaluating the projected changes in the 35-member ALLFORC ensembles. The range of the distribution is used to test the distinguishability of shifts in precipitation mean state projected by the ALLFORC and NATURAL ensembles: shifts *outside the range* are attributable to external (anthropogenic or natural) forcing and defined as distinguishable.

We choose 1950–1999 as the reference period for two reasons. First, the FLOR ensembles start from year 1941 and the first 10 years of simulation are treated as model adjustment to initial conditions; second, future changes relative to this recent past climate are more relevant for climate mitigation and adaptation (compared to changes relative to a farther past climate). In assessing the distinguishability of anthropogenic changes in future projections, we focus on the decadal time scale because decadal variations of precipitation (e.g., decadal drought) can impose drastic impacts on environment and society[1–3, 41].

**Sensitivity of distinguishability to ensemble size**. We examine the sensitivity of the distinguishability analysis to the ensemble size. The same distinguishability analysis is repeated for a subset of the two 35-member ALLFORC ensembles, including sets of 5, 10, 15, 20, 25, and 30 ensemble members (starting from the first 5 members, and then adding 5 members incrementally). The results are shown in Supplementary Figure 13. The area of distinguishable shifts in precipitation mean state during the same decade increases with the ensemble size (as expected); however, this increase gets weaker as the ensemble size grows, especially when it exceeds 25 members. This sensitivity test suggests that our distinguishability results using the 30- and 35-member ensembles are robust.

**Time of emergence**. The time of emergence of anthropogenic shifts in precipitation mean state is defined as the decade when projected precipitation changes, in the ALLFORC ensemble but not the NATURAL ensemble, first become distinguishable and remain so thereafter for at least two decades. Note that 2040s for FLOR and 2090s for CESM1 are not defined, since no projections are available after 2050 for FLOR and 2100 for CESM1.

**Code availability**. All relevant codes used in this work are available, upon request, from H.Z. (honghaiz@princeton.edu).

**Data availability**. The NCAR CESM1 data are available at http://www.cesm.ucar.edu/projects/community-projects/LENS/. All relevant data associated with the GFDL FLOR are available, upon request, from T.L.D. (tom.delworth@noaa.gov).

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

## Acknowledgements

We thank Dr. Jie He and Dr. Nathaniel Johnson for their insightful and constructive comments on the manuscript, Gabriel Vecchi and Karen Paffendorf for designing and conducting the FLOR ensembles, and the CESM Large Ensemble Community Project (http://www.cesm.ucar.edu/projects/community-projects/LENS/) and supercomputing resources provided by NSF/CISL/Yellowstone. H. Zhang is supported through Princeton University under block funding from NOAA/GFDL; T. Delworth is supported as a base activity of NOAA/GFDL.

## Author contributions

H.Z. designed the study, performed the analysis and wrote the initial paper; T.L.D. contributed in the interpretation and writing of the results.

## Additional information

**Competing interests:** The authors declare no competing interests.

