## [Peer Review File(PDF 2974 kb) · Nature Communications]

Reviewers' comments:

Reviewer #1 (Remarks to the Author):

The study investigates the emergence of precipitation changes regarding decadal internal variability, using several large model ensembles from two different models. It shows that some of the precipitation changes signals have already emerged and more are expected to do so in the future. The issue addressed is of crucial importance but I think what makes this study of particular interest is the use of large model ensembles, leading to robust and sensible results which are valuable for the community. Therefore I recommend that the study will be suitable for publication in Nature Communications after addressing a few comments.

The study is very clear, concise and well written. Everything is explained and nothing is hidden, which is appreciable. But it lacks links with the existing literature and existing results; that is what most of my comments are about. My only and main concern about the analysis itself is the relevance of dividing the year into two 6-months seasons. It is also a concern regarding the text because it makes a large part of it redundant. Some space could be saved and used to make more links with existing literature. Depending on whether this choice of seasons is decisive or not, it might require a modification of the analysis, this is why I prefer to recommend major revisions although this is rather a minor comment.

Major comment:

Could the authors explain their choice of dividing the year into two 6-months seasons? The conclusions are nearly identical and the part describing MJJASO results is mostly redundant with the NDJFMA part. Are the results sensitive to the choice of the seasons? This should be mentioned. If the results are sensitive to the choice of the seasons, perhaps the authors should reconsider it accordingly. I have especially in mind the Asian monsoon and I would be curious to know if more detectable changes can be detected from focusing more on the monsoon season.

Other comments:

Aerosol forcing can be strong and counteract the GhG forcing. It impacts the precipitation. I am wondering if the aerosols could slow down the emergence of precipitation detectable changes. Could the authors comment on that? (see e.g. Szopa et al. 2013 for the evolution of the aerosols radiative forcing). Could there be a relationship between the evolution of the percentage of detectable changes and the evolution of aerosol forcing?

The monsoon systems are not mentioned, but this would be interesting to comment the lack of emergence of the Asian monsoon changes by 2050 for example, and the fact that the signal finally emerges in the second half of the century (Fig. S6). Is it possible to link it with the compensating effects of aerosols and greenhouse gases (see e.g. Polson et al. 2014 for the decrease of the monsoon in response to aerosols, Guo et al. 2013 for other details on it)? Or is there a high internal variability for monsoon systems (or both)?

Line 121: I think the eastern subtropical oceans should be in the list of the regions where changes are already detectable in the 2000s. The authors could highlight that this is a fast emerging response and that this is consistent with He & Soden results that they mention previously line 113.

Line 119: the authors also point an expansion of the subtropical drying emergence during the following decades, do they think this could it be related to the slowest response of the Hadley cell expansion?

Line 197: when mentioning the stability of the pattern, the authors could refer to existing studies pointing this (e.g. Chavillaz et al. 2016).

About the percentages of Table S2, mentioned several times in the text: are they percentages of gridpoints or percentages of the globe area? My point is that high latitudes would have too much weight if not weighted by the gridpoints area.

Lines 225 and 228: 'tropical Pacific' should rather be 'equatorial Pacific'.

Line 231-232: 'The early emergence of anthropogenic warming in the tropics results primarily from the relatively weak internal variability in surface temperature'. Could the authors add a sentence clarifying why the emergence of the precipitation signal is different?

Figure 4: the numbers (associated with invisible contours I guess) appearing on the figures should be dropped.

References:

Szopa et al. 2013, doi: 10.1007/s00382-012-1408-y

Polson et al. 2014, doi: 10.1002/2014GL060811

Guo et al. 2013, doi: 10.5194/acp-13-1521-2013

Chavaillaz et al. 2016, doi: 10.1007/s00382-015-2882-9

Reviewer #2 (Remarks to the Author):

This manuscript uses multiple single-model large ensembles to examine the time of emergence of a signal of mean precipitation change. They also diagnose the sources of variability in these model simulations.

It seems to me that the authors have not made a compelling argument or articulation of how their work is novel. Specifically, Deser et al (2012) already touch on the time of emergence of precipitation, while He, Deser and Soden (2016) present an analysis of coupled versus AMIP simulations to attribute the sources of variability which seems to me to have substantial overlap.

Furthermore, the introduction attempts to frame this work as focusing on detection and attribution of decadal changes. This concept seems poorly posed to me, as it convolves forced and unforced changes. At a minimum, it needs to be articulated much better. But if this concept is going to form the basis of a new framework, then its utility should probably be demonstrated with some analysis.

Overall, I think it might be appropriate to flesh this work out more and then submit it to a more domain-oriented journal.

Specific issues

Line 44-6: I don't think detection and attribution of decadal changes is a well-posed problem, and thus I'm not convinced it is worth investigating. Your challenge is to convince readers (and reviewers) otherwise, or reframe the work.

Line 50-1: Citation of Deser et al 2012 that "anthropogenic impacts on precip internal variability are likely rather weak on decadal time scales". I'm not sure this is actually demonstrated by this work – it is certainly not their main point.

Line 60: "fine spatial resolution of the models (about 0.5~1 degree for atmosphere and land" – CESM1 is 1 degree resolution. 0.5 degrees is misleading. Also, this resolution is completely

standard at this point. There are versions of both CESM and GFDL run at 0.25 degrees.

Lines 245-7: Mean precipitation is not necessarily related to extreme precipitation – their changes are quite different.

Typos and smaller issues

Line 25: "a certain aspect"

Line 35: "Decadal changes" should be "Decadal variability" or "Decadal-duration variations"

Line 36: "decadal drought" should be "decade-long drought"

Lines 38 and 40: projecting should be predicting, especially on line 40

Line 44: suggests -> presents

Line 406: Kay et al was published in 2015, not 2014.

He, J., C. Deser, and B. J. Soden, 2016: Atmospheric and Oceanic Origins of Tropical Precipitation Variability. *J. Clim.*, doi:10.1175/JCLI-D-16-0714.1.

Response to Reviewers' comments:

Reviewer #1 (Remarks to the Author):

We thank the reviewer for his or her constructive and detailed comments that help to solve some overlooked issues and to improve the manuscript. In particular, we have repeated our analysis with the annual mean precipitation (and PmE) and found overall similar results; therefore, we have replaced the original seasonal results with the new annual mean results in the main text and moved the seasonal results to the supplements. We really appreciate this comment that helps to make our paper more succinct. We have also cited the relevant existing literature that we had overlooked. Below are our detailed responses.

The study investigates the emergence of precipitation changes regarding decadal internal variability, using several large model ensembles from two different models. It shows that some of the precipitation changes signals have already emerged and more are expected to do so in the future. The issue addressed is of crucial importance but I think what makes this study of particular interest is the use of large model ensembles, leading to robust and sensible results which are valuable for the community. Therefore I recommend that the study will be suitable for publication in Nature Communications after addressing a few comments.

The study is very clear, concise and well written. Everything is explained and nothing is hidden, which is appreciable. But it lacks links with the existing literature and existing results; that is what most of my comments are about. My only and main concern about the analysis itself is the relevance of dividing the year into two 6-months seasons. It is also a concern regarding the text because it makes a large part of it redundant. Some space could be saved and used to make more links with existing literature. Depending on whether this choice of seasons is decisive or not, it might require a modification of the analysis, this is why I prefer to recommend major revisions although this is rather a minor comment.

Major comment:

Could the authors explain their choice of dividing the year into two 6-months seasons? The conclusions are nearly identical and the part describing MJJASO results is mostly redundant with the NDJFMA part. Are the results sensitive to the choice of the seasons? This should be mentioned. If the results are sensitive to the choice of the seasons, perhaps the authors should reconsider it accordingly. I have especially in mind the Asian monsoon and I would be curious to know if more detectable changes can be detected from focusing more on the monsoon season.

We thank the reviewer for this comment. We have repeated our analysis with annual mean precipitation and found overall similar results to the 6-month seasonal mean results. We have therefore moved the seasonal mean results to the supplements and added to the main text the new annual mean results along with discussions about the important differences between the annual and seasonal mean results. This has led to substantial revisions of the main text (see the Results section), although our conclusions remain essentially unchanged.

The MJJASO period corresponds roughly to the Asian monsoon season (May to September, Polson et al. 2014), and indeed, the detectable precipitation changes emerge earlier compared to the NDJFMA season (see more responses to your comment below on the monsoon systems). This has been discussed in the paper (see L192-204).

Other comments:

Aerosol forcing can be strong and counteract the GHG forcing. It impacts the precipitation. I am wondering if the aerosols could slow down the emergence of precipitation detectable changes. Could the authors comment on that? (see e.g. Szopa et al. 2013 for the evolution of the aerosols radiative forcing). Could there be a relationship between the evolution of the percentage of detectable changes and the evolution of aerosol forcing?

We agree with the reviewer that aerosol forcing has an overall cooling effect and can partly counteract the GHG warming. But in the context of relative changes between future and historical periods, for example, [2000-2050] relative to [1950-1999] as is the focus of our work, the effect of aerosol forcing on future climate relative to past climate is probably a weak “warming”. As concluded in Szopa et al. (2013), “According to the RCP scenarios, the aerosol content, after peaking around 2010, is projected to decline strongly and monotonically during the twenty-first century for the RCP8.5, 4.5 and 2.6 scenarios”. In other words, the projected aerosol cooling will be weaker in the future (after the 2010s) than in the past (since the 1950s when anthropogenic aerosols rapidly increased) and the relative effect on climate is a future “warming”. Therefore, the future strong decline in aerosol forcing in the RCP4.5 and 8.5 scenarios used here should not slow down (but facilitate) the emergence of a GHG-induced precipitation signal (after the time when the aerosol content decreases below the 1950-1999 mean).

Our experiments, however, cannot separate the impacts of aerosol and GHG forcing on the evolution of detectable precipitation changes, because both forcing is included in the same ALLFORC ensemble but both excluded in the NATURAL ensemble (except for volcanic aerosols). Nonetheless, our results present the evolution of detectable precipitation changes in response to the **overall** anthropogenic forcing projected in the RCP4.5 and 8.5 emission scenarios, and provide information that is directly relevant to climate mitigation and adaptation.

The monsoon systems are not mentioned, but this would be interesting to comment the lack of emergence of the Asian monsoon changes by 2050 for example, and the fact that the signal finally emerges in the second half of the century (Fig. S6). Is it possible to link it with the compensating effects of aerosols and greenhouse gases (see e.g. Polson et al. 2014 for the decrease of the monsoon in response to aerosols, Guo et al. 2013 for other details on it)? Or is there a high internal variability for monsoon systems (or both)?

Precipitation changes over the Asian monsoon region, particularly southeast Asia and parts of India, are actually detectable around 2020-2040 in FLOR and around 2040s in CESM1 during MJJASO, roughly the monsoon season (see Fig. S5 in the revised paper). This earlier detectability in FLOR is due to its stronger signal (moistening), since its internal variability is also stronger over these regions during MJJASO (see Fig. S5 and S7) compared to CESM1. The weaker signal in CESM1 seems to be associated with the counteraction between aerosols and GHGs before 2040s, because CESM1 projects an ensemble mean drying (presumably due to aerosols, Polson et al. 2014) that is detectable during 2000-2020 when the aerosol content is around the highest (Szopa et al., 2013) and an ensemble mean moistening (due to GHGs, Polson et al. 2014) after 2030s (see the figure below). In contrast, FLOR does not project this ensemble mean drying-to-moistening transition, but just a moistening signal growing with time. This difference between FLOR and CESM1 over the monsoon regions is likely associated with their different sensitivities to aerosol/GHG forcing (which is beyond the scope of this study). This discussion has been added to the paper as one of the differences between the annual and seasonal detectability (see L192-204).

Decadal evolution of changes in MJJASO precipitation mean state relative to 1950~1999 during 2000~2050. (a) CESM1 ALLFORC RCP8.5 (35 members), (b) FLOR ALLFORC RCP4.5 (35 members) and (c) FLOR NATURAL (30 members) ensemble average. Time goes down, as denoted to the left of the figure.

Line 121: I think the eastern subtropical oceans should be in the list of the regions where changes are already detectable in the 2000s. The authors could highlight that this is a fast emerging response and that this is consistent with He & Soden results that they mention previously line 113.

The eastern subtropical oceans have been added to the list (see L122).

Line 119: the authors also point an expansion of the subtropical drying emergence during the following decades, do they think this could it be related to the slowest response of the Hadley cell expansion?

We think that the expansion of the subtropical drying emergence after the 2010s is related to both the fast (He and Soden, 2017) and slow (i.e., Hadley cell expansion) responses, since the radiative forcing varies with time. The poleward expansion of Hadley Cells during NDJFMA has been documented in our previous work (Zhang et al., 2016) using a similar set of GFDL model ensembles conducted with FLOR and CM2.5 (CM2.5 is the same as the FLOR model used here except for a higher spatial resolution in its ocean/sea ice components and no flux adjustment).

Zhang, et al., 2016: Detection, Attribution, and Projection of Regional Rainfall Changes on (Multi-) Decadal Time Scales: A Focus on Southeastern South America. *J. Clim.*, 29, 8515–8534, doi:10.1175/JCLI-D-16-0287.1.

Line 197: when mentioning the stability of the pattern, the authors could refer to existing studies pointing this (e.g. Chavaillaz et al. 2016).

Chavaillaz et al. 2016 has been cited (see L227).

About the percentages of Table S2, mentioned several times in the text: are they percentages of gridpoints or percentages of the globe area? My point is that high latitudes would have too much weight if not weighted by the gridpoints area.

We appreciate this comment that helps with clarification. All the percentages in Table S2 are area-weighted, so they are the percentages of the global area. This has been highlighted in the captions of Table S2 and a new Table 1.

Lines 225 and 228: 'tropical Pacific' should rather be 'equatorial Pacific'.

Revised (see L255 and 258).

Line 231-232: 'The early emergence of anthropogenic warming in the tropics results primarily from the relatively weak internal variability in surface temperature'. Could the authors add a sentence clarifying why the emergence of the precipitation signal is different?

Revised to "The early emergence of anthropogenic warming in the tropics results primarily from the relatively weak internal variability in surface temperature, while the early emergence of anthropogenic moistening there arises mainly from the relatively strong signal (since the associated internal variability is strong)". See L262-265.

Figure 4: the numbers (associated with invisible contours I guess) appearing on the figures should be dropped.

We thank the reviewer for the detailed comment. The figure (now Fig. 5) has been replotted without the numbers and with a different color scale to highlight the main differences among the control simulations.

References:

Szopa et al. 2013, doi: 10.1007/s00382-012-1408-y
Polson et al. 2014, doi: 10.1002/2014GL060811
Guo et al. 2013, doi: 10.5194/acp-13-1521-2013
Chavaillaz et al. 2016, doi: 10.1007/s00382-015-2882-9

Reviewer #2 (Remarks to the Author):

We thank the reviewer for his or her comments that help to solve some overlooked issues.

This manuscript uses multiple single-model large ensembles to examine the time of emergence of a signal of mean precipitation change. They also diagnose the sources of variability in these model simulations.

It seems to me that the authors have not made a compelling argument or articulation of how their work is novel. Specifically, Deser et al (2012) already touch on the time of emergence of precipitation, while He, Deser and Soden (2016) present an analysis of coupled versus AMIP simulations to attribute the sources of variability which seems to me to have substantial overlap.

The novelty of our work lies in the detectability of anthropogenically forced decadal precipitation changes against unpredictable internal climate variability. Many recent studies have shown that unpredictable internal climate variability dominates precipitation changes on regional and decadal scales, which suggests little hope in the detection of the associated anthropogenic component. Here we use multiple large model ensembles of climate change experiments to demonstrate that, even on regional and decadal scales, anthropogenic shifts in precipitation mean state can be readily detected against the dominant unpredictable internal climate variability, and quantitatively show that anthropogenic forcing is projected to cause detectable shifts in precipitation mean state within the next few decades over the majority of the planet. This is a new finding on the detectability of anthropogenic precipitation changes on decadal time scale that to our knowledge has not been reported before. It is essential to use large ensembles to make this finding, and the use of large ensembles with multiple models strengthens our conclusions.

Although Deser et al. (2012) (D12 hereafter) touched on the time of emergence of precipitation, they focused on the role of the dominant unpredictable internal climate variability in masking out the anthropogenic signal, implying that the anthropogenic signal in precipitation changes is hardly detectable on multi-decadal and shorter time scales. However, we here have demonstrated that a certain aspect of anthropogenic changes in precipitation (i.e., shifts in mean state) is readily detectable even on regional and decadal scales. (Many other differences exist. For example, D12's emergence analysis was based on a statistical analysis (student-t test), while our detectability analysis is based on the physical comparison between anthropogenic signal (i.e., ensemble average) and internal climate variability (assessed from long control simulations)—which is an attribution analysis; D12 used one large ensemble conducted with NCAR's older generation model CCSM3, while we use three state-of-the-art large ensembles conducted with NCAR's CESM1 and GFDL's high resolution FLOR.)

The source of precipitation internal variability is not our focus (as stated above), but just a side point. Nonetheless, the differences between our study and He et al. (2016) are apparent: He et al. investigated the origin of precipitation internal variability *only in tropics*, while we examined the entire globe. The extratropical part is not investigated by He et al. (2016). As we demonstrated here, the sources of precipitation internal variability on decadal and longer time scales are very different between tropics (ocean-atmosphere coupling contributes substantially) and extratropics (atmosphere/land internal dynamics dominate).

Furthermore, the introduction attempts to frame this work as focusing on detection and attribution of decadal changes. This concept seems poorly posed to me, as it convolves forced and unforced changes. At a minimum, it needs to be articulated much better. But if this concept is going to form the basis of a new framework, then its utility should probably be demonstrated with some analysis.

The Reviewer summarized our work by “This manuscript uses multiple single-model large ensembles to examine the time of emergence of a signal of mean precipitation change”. In fact, our work focused on the detectability of anthropogenic decadal precipitation changes mostly in climate projections (after 2000) against natural variability, which essentially falls into the category of detection and attribution problem. Because our detectability analysis is conducted on decadal time scales (i.e., for each decade after 2000), this decadal evolution of the detectability is naturally summarized by the time of emergence (ToE) as well as the fraction of the global area with detectable signal. Thus, the time of emergence, along with the fraction of global area, is merely a metric we used to illustrate the detectability analysis, and **our conclusions would remain the same even if the ToE part was cut out**. Our work directly addressed and focused on the detectability of anthropogenic decadal precipitation changes in the 21st-century climate projections against natural variability, and should be framed as a detection and attribution problem.

In addition, we argue that the detection and attribution problem underlies the time of emergence. In order to claim that a signal emerges or will emerge, the signal has to be first detected against “noise” (that is, a change is attributable between signal and noise). Therefore, the ToE itself is essentially a detection and attribution problem, although the detection and attribution problem need not always invoke the concept of ToE.

We appreciate the reviewer focusing on this issue, and we have revised the introduction to better articulate our motivation (see more responses below to the first specific comment).

Editor communication: In addition, the reviewer has also clarified that, in their opinion, as well as conflating time of emergence and D&A, the study conflates mean precipitation and decadal-averaged precipitation. They disagree with the assumption that decadal-average precipitation represents the mean state, which cannot be because of the large internal variability, even at decadal timescales.

Our detectability analysis on decadal time scales is enabled by the large model ensembles. The mean state for each decade is averaged over 350 (300) model years for the 35- (30-) member ensemble. This large number of model years for each decade allows a robust assessment of the mean state for that decade. This is actually one of novelties of our work: using multiple state-of-the-art large model ensembles to assess the detectability of anthropogenic changes in precipitation mean state on decadal time scales.

Overall, I think it might be appropriate to flesh this work out more and then submit it to a more domain-oriented journal.

Specific issues

Line 44-6: I don't think detection and attribution of decadal changes is a well-posed problem, and thus I'm not convinced it is worth investigating. Your challenge is to convince readers (and reviewers) otherwise, or reframe the work.

We appreciate the reviewer's comment, and we hope that our revised version can make this case much more clearly and convincingly. Our work specifically addresses the detectability of decadal anthropogenic precipitation changes against internal climate variability. It falls into the category of detection and attribution: the detection of an anthropogenically forced signal against unpredictable internal climate variability, and the attribution between anthropogenic forcing and internal climate dynamics in terms of their contribution to decadal precipitation changes. We have revised the opening two paragraph (see L35-48) to articulate this point by the following logic:

1. Precipitation decadal changes can drastically impact environment and society; predicting these changes has been a long-term goal for scientists and the public;
2. Precipitation decadal changes are dominated by internal climate variability that is inherently unpredictable beyond a decade;
3. This dominance of unpredictable internal climate variability suggests an enormous challenge for the detection of anthropogenically caused precipitation decadal changes, owing to the weak "signal-to-noise" ratio;
4. Detecting anthropogenic precipitation decadal changes in climate projections can both improve understanding and provide information that can guide climate mitigation and adaptation.

In the second paragraph, we have also rephrased the main question addressed in this work and deleted the 2nd question on ToE in the previous version that might have caused the confusion. See L49-54.

Line 50-1: Citation of Deser et al 2012 that "anthropogenic impacts on precip internal variability are likely rather weak on decadal time scales". I'm not sure this is actually demonstrated by this work – it is certainly not their main point.

This statement was based on Fig. 13 (see below) in D12. We agree that it is not a conclusive point that was explicitly stated in D12, but it seems that changes in the shape of the PDFs (i.e., internal variability) are weaker compared to the shifts of the PDFs (i.e., mean state). However, since this statement is not essential to our argument, we have deleted it to avoid potential confusions. See L50-52.

Figure 13 from D12. "Histograms of regionally-averaged trends over the Eurasian–North Atlantic sector in DJF for SLP (left), Precip (middle) and TS (right) in DJF from the 40-member CCSM3 ensemble (open red bars) and the "178-member" CAM3 control integration (grey filled bars). Trends are computed over the period 2005–2060 for CCSM3 and for 56-year non-overlapping segments for CAM3. For all panels, the x axis is in units of standard deviation based on CAM3, and the y axis is in units of the number of ensemble members divided by the total number of ensemble members"

Line 60: "fine spatial resolution of the models (about 0.5~1 degree for atmosphere and land" – CESM1 is 1 degree resolution. 0.5 degrees is misleading. Also, this resolution is completely standard at this point. There

are versions of both CESM and GFDL run at 0.25 degrees.

We thank the reviewer for pointing out this confusion. 0.5 degree refers to the atmosphere/land components of the GFDL FLOR. This has been revised (see L59-60).

We agree that there are global climate models that run at 0.25 degrees or even higher resolution, but these high-resolution global models are normally too expensive to run in the large-ensemble configuration. Our two FLOR ensembles used here are novel in their combination of relatively high spatial resolution and large ensemble size.

Lines 245-7: Mean precipitation is not necessarily related to extreme precipitation – their changes are quite different.

We agree with the reviewer, but we only meant that anthropogenic shifts in precipitation mean state will change the background in which hydrological extremes occur, and we didn't talk about the changes in precipitation extremes themselves. This has been revised (see L277-280).

Typos and smaller issues

Line 25: "a certain aspect"
Revised (see L25)

Line 35: "Decadal changes" should be "Decadal variability" or "Decadal-duration variations"
Revised (see L35)

Line 36: "decadal drought" should be "decade-long drought"
Revised (see L36)

Lines 38 and 40: projecting should be predicting, especially on line 40
"Projecting" on L38 is what we meant: simulating climate changes in response to anthropogenic forcing, which is a boundary-condition problem, while "predicting" is an initial-condition problem. L40 has been deleted.

Line 44: suggests -> presents
Revised (see L43)

Line 406: Kay et al was published in 2015, not 2014.
Revised (see L437)

He, J., C. Deser, and B. J. Soden, 2016: Atmospheric and Oceanic Origins of Tropical Precipitation Variability. J. Clim., doi:10.1175/JCLI-D-16-0714.1.

Reviewers' comments:

Reviewer #1 (Remarks to the Author):

The authors have addressed most of my comments, however there are still some answers which are not entirely satisfying in my opinion.

For example I don't think I agree with their answer to my second comment about aerosols (and I don't think it is consistent with their answer to my following comment about the aerosols and the monsoon systems), but since they mentioned in the manuscript the effect of aerosols on the monsoon, I will not ask for more and I will not discuss it further.

My actual concern is that they only partly answered my comment about the dependence of the results on the choice of the seasons. When I asked if more detectable changes on the Asian monsoon could be detected from focusing more on the monsoon season, I meant by picking JJA for example, or even looking month by month (of course I know that MJJASON encompasses the Asian monsoon season...). That being said, I understand this is not a main issue. Therefore, if the other reviewer and the editor are happy with the manuscript as it is, I will consider this concern as minor and not standing in the way of publication, thus recommending the manuscript for publication in Nature Communications (as it is). But if the manuscript comes back in revisions, I think this point is worth being clarified.

Reviewer #2 (Remarks to the Author):

I appreciate the authors' efforts to address my comments. Unfortunately, they have not addressed my primary concern, which is that they consider decadal average precipitation to represent the mean state. There is substantial variability in precipitation on decadal timescales, for example, from phenomena including the IPO, and demonstrated by the recent hiatus period – which the authors state in the first sentence of the abstract. Because of this variability, decadal average precipitation cannot be said to represent the mean state. This has a number of implications; for example, the discussion of background literature connecting changes in mean precipitation to their decadal averages of precipitation is not meaningful. As a result, I cannot recommend this manuscript for publication as long as it identifies decadal average precipitation with the mean state.

Response to Reviewers' comments:

Reviewer #1 (Remarks to the Author):

The authors have addressed most of my comments, however there are still some answers which are not entirely satisfying in my opinion.

We thank the reviewer for his/her further valuable and constructive comments. As to the dependence of the results on the choice of seasons, we have further repeated our analysis for JJA and DJF seasons, and found similar and consistent results. Detailed responses are included below. Minor revisions are added to the paper.

For example I don't think I agree with their answer to my second comment about aerosols (and I don't think it is consistent with their answer to my following comment about the aerosols and the monsoon systems), but since they mentioned in the manuscript the effect of aerosols on the monsoon, I will not ask for more and I will not discuss it further.

My actual concern is that they only partly answered my comment about the dependence of the results on the choice of the seasons. When I asked if more detectable changes on the Asian monsoon could be detected from focusing more on the monsoon season, I meant by picking JJA for example, or even looking month by month (of course I know that MJJASON encompasses the Asian monsoon season...). That being said, I understand this is not a main issue. Therefore, if the other reviewer and the editor are happy with the manuscript as it is, I will consider this concern as minor and not standing in the way of publication, thus recommending the manuscript for publication in Nature Communications (as it is). But if the manuscript comes back in revisions, I think this point is worth being clarified.

We have repeated our detectability analysis for JJA and DJF and compared the new results with the annual, NDJFMA and MJJASO results. We found that

- (1) Overall, the detectability is similar for annual, JJA, DJF, MJJASO and NDJFMA.
- (2) The differences between annual and JJA (DJF) means are consistent with those between annual and MJJASO (NDJFMA) means that have been discussed in the paper, and are also consistent with those between the 3-month JJA (DJF) and 6-month MJJASO (NDJFMA) means.

Please see the figures below and the associated discussions. We have revised our paper to include these discussions (see Lines 97-103; 178; 190-192; 201; 203).

Figure 1. Fraction of global area with detectable changes in precipitation mean state for annual (black) and various seasonal means (see legends). (a) FLOR ALLFORC, (b) CESM1 ALLFORC.

The fraction and its increase are overall similar for annual and seasonal means. The main difference is that the annual fraction is slightly larger than the 6-month seasonal fractions and the 6-month seasonal fractions are slightly larger than the 3-month seasonal fractions. That is, the shorter the average time, the smaller the fraction and the weaker the detectability. The difference between annual and MJJASO/NDJFMA has been attributed to the weaker noise for the annual mean than the seasonal means in the first-round revision. Here the difference between annual and JJA/DJF and between MJJASO/NDJFMA and JJA/DJF can also be explained by the difference in the noise amplitude. The 3-month JJA/DJF means have the strongest noise compared to annual and 6-month means (Fig. 2 and 3) and thus have the weakest signal-to-noise ratio (compare Fig. 6 and 7 here with Fig. 3, S6-7 in the paper) and detectability.

Figure 2. Ratio of the average between JJA and DJF noise to the annual noise.

Figure 3. Ratio of the average between MJJASO and NDJFMA noise to the annual noise. (the same as Fig. 4 in the paper).

Figure 4. The same as Fig. 1 in the paper but for DJF.

Note the monsoonal regions: during DJF, south Asia shows weak or no detectability by 2050, but southeastern South America shows early detectability in both models, consistent with results for NDJFMA.

Figure 5. The same as Fig. 1 in the paper but for JJA.

Note the monsoonal regions: during JJA, south Asia shows early detectability by 2050 (earlier in FLOR than CESM1), but southeastern South America shows weak or no detectability, consistent with results for MJJASO.

Figure 6. Signal to noise ratio (top two rows) and noise (bottom row). The same as Fig. 3, S6 and S7 in the paper but for DJF.

DJF has the strongest noise and weakest S/N ratio compared to annual and NDJFMA.

Figure 7. Signal to noise ratio (top two rows) and noise (bottom row). The same as Fig. 6 but for JJA
 JJA has the strongest noise and weakest S/N ratio compared to annual and MJJASO.

Reviewer#2 (Remarks to the Author):

Thank you for your further valuable comments on our paper. We realized that you had raised this question in the last round review. We tried to provide an answer to your question, but our response was probably not as clear as it could have been. So, we now try to provide a more detailed explanation to your comments, and hopefully this will clarify the issue.

New comments: “I appreciate the authors’ efforts to address my comments. Unfortunately, they have not addressed my primary concern, which is that they consider decadal average precipitation to represent the mean state. There is substantial variability in precipitation on decadal timescales, for example, from phenomena including the IPO, and demonstrated by the recent hiatus period – which the authors state in the first sentence of the abstract. Because of this variability, decadal average precipitation cannot be said to represent the mean state. This has a number of implications; for example, the discussion of background literature connecting changes in mean precipitation to their decadal averages of precipitation is not meaningful. As a result, I cannot recommend this manuscript for publication as long as it identifies decadal average precipitation with the mean state.”

Previous comments: “the study conflates mean precipitation and decadal-averaged precipitation. They (the reviewer) disagree with the assumption that decadal-average precipitation represents the mean state, which cannot be because of the large internal variability, even at decadal timescales.”

We agree that it would be very uncertain to define a mean state with just 10 years of data (decadal average) due to the large role of low-frequency internal variability (as implied by Fig. S13 in the supplements). Consistent with the issue that you raised, we did not compute a decadal mean state from only 10 points. Rather, in our study, the decadal average is calculated using 10 years from each of 35 (or 30) ensemble members, so that each decadal mean is the average of 350 (or 300) model years - not simply the mean of 10 years. Because all ensemble members by design are equally possible but different realizations of the climate system, averaging across all ensemble members for each decade (350 or 300 individual years) allows a much more refined assessment of the mean state for that decade. This process averages across random phases of various types of internal variability, such as the IPO, and therefore provides a more robust estimate of the mean. We are also very aware that there is still uncertainty in the mean state even defined with such a larger number of model years. That is why we further conducted the detectability analysis—to assess the robustness of decadal changes in the mean state (calculated as the mean over all ensemble members) against what could occur purely from low-frequency internal climate variability. Please note that this analysis is not some simple statistical test as often used in literature, but is more physical in the sense that it is an attribution between forced changes and internal variability. In other words, **a more robust estimate of the decadal mean state is enabled by the large model ensembles in our study.** This is one of the novelties of our work, as summarized by Reviewer #1 in his/her first-round comments: “The issue addressed is of crucial importance but I think what makes this study of particular interest is the use of large model ensembles, leading to robust and sensible results which are valuable for the community”.

The text was not as clear as it could have been on this point, thereby contributing to a lack of clarity on exactly what we meant by a “decadal average”. This is a point that should certainly be

communicated more clearly with a revision to the text, which will articulate how the mean state is defined using the large model ensembles, and what the implications of this process are. We have revised the text accordingly to further clarify this point (See Lines 73-76).

Original text: “For each decade after 2000, the mean state is defined as the ensemble average within each large model ensemble (the average is calculated over $10 \cdot N$ model years, where N is the ensemble size). Three large ensembles are analyzed...”

Revised text: “For each decade after 2000, the mean state is defined as the ensemble average within each large model ensemble over $10 \cdot N$ model years, where N is the ensemble size (10 years from each ensemble member). For example, the decadal mean state for 2030-2039 is calculated as the numerical average over 350 model years in a 35-member ensemble. The relatively large model ensembles allow a more robust estimate of the mean state for each decade. Three large ensembles are analyzed...”

REVIEWERS' COMMENTS:

Reviewer #1 (Remarks to the Author):

The authors have addressed my comments and answered my questions. I now recommend the manuscript for publication in Nature Communication.

Reviewer #2 (Remarks to the Author):

I would like to thank the authors for their revisions to the text regarding differentiating between decadal average precipitation and mean precipitation, and for their patient responses to my comments. I now understand that by the mean state, they meant the ensemble-mean decadal average precipitation.

Since the focus on the study is on changes of ensemble-mean decadal averaged precipitation, it seems that the word "detectable" is not appropriate to describe this analysis. One important example is the last sentence of the abstract is "Our findings imply detectable anthropogenic shifts in precipitation mean state over the majority of the planet within the next few decades." An average over 30 ensemble members will never be detectable, because we only have one observable realization of the climate system. So, the word "detectability" should be removed from the manuscript, and replaced with a different word, such as "forced."

Specific comments

Title: "Anthropogenic" should be "Anthropogenically Forced"

Line 28: "are already" -> "should already be"

Line 31-33: I think this needs to be qualified as detectability, because it is only hypothetical – the mean of a 30 member ensemble will never actual be detectable. "detectable... within the next few decades" implies that it is something we will be able to observe. But we will not, because we will still only have one realization of the observable climate. It might be that the word

Line 51: Make this "future ensemble projections" to make it clear that you're not conflating a simple decadal average of precipitation (eg in observations) with an estimate much closer to the statistical mean precipitation derived from an ensemble.

Line 90: Shouldn't the detection and attribution steps be different, making use of the single-forcing ensembles for the attribution step?

Line 240: Figures 2 and 5 are the only relevant figures here, as far as I can tell.

Line 243-5: I think it's an overinterpretation to say that atmosphere/land internal dynamics are key over most of the globe. A more appropriate statement might be, "...precipitation low-frequency internal variability arises primarily from internal dynamics of the atmosphere and land-atmosphere interactions over most land areas" or "some of the globe"

Line 250: The difference between slab ocean and fully coupled simulations goes beyond momentum exchange. Interactions between circulation and heat exchange are probably also important in some cases – see Armour et al 2016 for an example.

Line 253: "atmosphere/land internal dynamics" should be atmospheric internal dynamics and land-atmosphere interactions

Line 283: "anthropogenic changes" should be "anthropogenically-forced changes"

Line 353-4: Linear interpolation is an inappropriate regridding technique for precipitation. Please employ a conservative regridding technique instead, e.g. Jones (1999).

Armour, K. C., J. Marshall, J. R. Scott, A. Donohoe, and E. R. Newsom, 2016: Southern Ocean warming delayed by circumpolar upwelling and equatorward transport. *Nat. Geosci.*, 9, 549–554, doi:10.1038/ngeo2731.

Jones, P. W., 1999: First- and Second-Order Conservative Remapping Schemes for Grids in Spherical Coordinates. *Mon. Weather Rev.*, 127, 2204–2210, doi:10.1175/1520-0493(1999)127<2204:FASOCR>2.0.CO;2.

Response to Reviewers' comments:

Reviewer #2 (Remarks to the Author):

We thank the reviewer for his/her further detailed valuable and constructive comments and have revised our manuscript accordingly. Below are our point-to-point responses to the comments.

I would like to thank the authors for their revisions to the text regarding differentiating between decadal average precipitation and mean precipitation, and for their patient responses to my comments. I now understand that by the mean state, they meant the ensemble-mean decadal average precipitation.

Since the focus on the study is on changes of ensemble-mean decadal averaged precipitation, it seems that the word "detectable" is not appropriate to describe this analysis. One important example is the last sentence of the abstract is "Our findings imply detectable anthropogenic shifts in precipitation mean state over the majority of the planet within the next few decades." An average over 30 ensemble members will never be detectable, because we only have one observable realization of the climate system. So, the word "detectability" should be removed from the manuscript, and replaced with a different word, such as "forced."

Thank you for pointing out this issue. In order to avoid confusion, we now define more explicitly, and earlier in the manuscript, what we call "detectable" (See L25-27; L53-55). By 'detectable', we mean that a decadal shift in precipitation **mean state** can be distinguished from (i.e., detected against) what could occur from random internal climate variability with time scales longer than a decade (such as IPO, AMO, etc.), and therefore can be attributed to external forcing.

The shift in mean state is one aspect (component) of climate change. Other aspects of climate change include contributions from random internal variability itself and changes in internal variability caused by external forcing. In this work, we only focus on the shift in mean state forced by external forcing, and assess its detectability against random internal variability. As in our previous responses, we agree that estimates of decadal mean state with a single-member realization are very uncertain, which is why we use large model ensembles that allow a more robust estimate of mean state on decadal time scales. Despite this difference between a single realization and large ensembles, the concept of detectability is the same: whether a signal—either the shift in mean state or total changes in response to external forcing—can be distinguished from what could occur from random internal variability.

Specific comments

Title: "Anthropogenic" should be "Anthropogenically Forced"

Revised. See L2

Line 28: "are already" -> "should already be"

We appreciate the suggestion. Since the results stated here refer explicitly to model simulations, we think it's appropriate to say ... "are already" detectable in our models.

Line 31-33: I think this needs to be qualified as detectability, because it is only hypothetical – the mean of a 30 member ensemble will never actual be detectable. "detectable... within the next few decades" implies that it is something we will be able to observe. But we will not, because we will still only have one realization of the observable climate. It might be that the word

Please see our response to the overall comments above.

Line 51: Make this "future ensemble projections" to make it clear that you're not conflating a simple decadal average of precipitation (eg in observations) with an estimate much closer to the statistical mean precipitation derived from an ensemble.

Thanks for the suggestion that helps to improve the clarity. Revised. See L52

Line 90: Shouldn't the detection and attribution steps be different, making use of the single-forcing ensembles for the attribution step?

Thank you for bringing up this point. The attribution step normally requires multiple single-forcing ensembles (such as greenhouse gas-only, aerosol-only, ozone-only, etc.), but the detection and attribution steps can be the same. In our work, the detectability analysis is an attribution between externally forced changes and internally generated variability. Therefore, if a change is detectable against internal climate variability, it is also attributable to external forcing. The comparison between the two ensembles with and without anthropogenic forcing (i.e., ALLFORC and NATURAL) is a further attribution between anthropogenic and natural (i.e., solar and volcanic) forcing. We have revised the text to clarify this point. See L53-59

Line 240: Figures 2 and 5 are the only relevant figures here, as far as I can tell.

Thank you for pointing this out. The figure citation here was not as clear as it could have been. The column (a) in Figures 1 and S3-5 shows the detectability analysis in CESM1 using both its fully coupled (denoted by gray crosses) and atmosphere/land-only (blue stippling) control simulations, and the results are similar. Both Figures 2 and 5 are further derived from Figure 1 in the main text. We have revised the figure citation to clarify it. See L256

Line 243-5: I think it's an overinterpretation to say that atmosphere/land internal dynamics are key over most of the globe. A more appropriate statement might be, "...precipitation low-frequency internal variability arises primarily from internal dynamics of the atmosphere and land-atmosphere interactions over most land areas" or "some of the globe"

We appreciate the comment. It's been revised to "... over most land areas and middle-to-high latitude oceans the variance of precipitation low-frequency internal variability arises primarily from internal dynamics of the atmosphere and land-atmosphere interactions, while in the tropics and subtropics ocean dynamics contribute substantially...". See L259-263

Line 250: The difference between slab ocean and fully coupled simulations goes beyond momentum exchange. Interactions between circulation and heat exchange are probably also important in some cases – see Armour et al 2016 for an example.

We agree with the reviewer’s argument. This has been revised to “...ocean dynamics in the tropics and subtropics can amplify precipitation low-frequency internal variability mainly through ocean-atmosphere thermodynamic coupling (i.e., heat and moisture exchange), but in the equatorial Pacific, ocean-atmosphere dynamical coupling (i.e., momentum exchange) is also required.”. See L264-268

Line 253: “atmosphere/land internal dynamics” should be atmospheric internal dynamics and land-atmosphere interactions

Thanks for this suggestion that improves the clarity. Revised. See L261, 270

Line 283: “anthropogenic changes” should be “anthropogenically-forced changes”

Revised. See L301

Line 353-4: Linear interpolation is an inappropriate regridding technique for precipitation. Please employ a conservative regridding technique instead, e.g. Jones (1999).

We have tried a globally conservative remapping method (https://www.ncl.ucar.edu/Document/Functions/Built-in/area_conserve_remap.shtml) and found nearly identical results. The text has been revised accordingly. See L371-372

For your reference, the figure below compares the results based on the original linear method and the new conservative method for annual precipitation using CRU observations.

Original results (Fig. S1a-b)

Model performance in precipitation change [1996~2005]-[1950~1995], against CRU v3.24.01

New results:

(a) FLOR, Annual

(b) CESM1, Annual